# Direct Oral Anticoagulants for the Treatment of Unusual-Site Venous Thrombosis: An Update

**DOI:** 10.3390/pharmaceutics17030342

**Published:** 2025-03-07

**Authors:** Anabel Franco-Moreno, Elena Madroñal-Cerezo, Ana Martínez-Casa-Muñoz, Judith Ortiz-Sánchez, Cristina Lucía Ancos-Aracil

**Affiliations:** 1Venous Thromboembolism Unit, Department of Internal Medicine, Hospital Universitario Infanta Leonor, Avenida Gran Via del Este, 80, 28031 Madrid, Spain; 2Venous Thromboembolism Unit, Department of Internal Medicine, Hospital Universitario de Fuenlabrada, Camino del Molino, 2, Fuenlabrada, 28942 Madrid, Spain; 3Hospital Universitario Gregorio Marañón, Calle del Dr. Esquerdo, 46, 28007 Madrid, Spain; 4Venous Thromboembolism Unit, Hospital Universitario de Torrejón, Calle Mateo Inurria, Torrejón de Ardoz, 28850 Madrid, Spain

**Keywords:** direct oral anticoagulants, randomized clinical trials, review, unusual-site thrombosis, venous thrombosis

## Abstract

Direct oral anticoagulants (DOACs) have emerged as the preferred oral anticoagulant therapy for patients with deep vein thrombosis of the lower extremities and pulmonary embolism. DOACs offer several advantages over vitamin K antagonists, including fixed dosage, fewer drug interactions, faster onset of action, and a lower risk of major bleeding, especially intracranial. Although evidence on the use of DOACs in unusual-site venous thrombosis (USVT) is limited, their use in such cases is becoming increasingly common. This narrative review examines the evidence derived from randomized controlled trials, and large observational studies focused on the use of the DOACs in USVT, including cerebral, splanchnic, upper extremity, ovarian, renal, and retinal vein thrombosis. In addition, it also provides practical advice for their use in these clinical settings according to the updated scientific literature.

## 1. Introduction

Unusual-site venous thrombosis (USVT) refers to thrombosis occurring in sites outside the veins of the lower limbs and the pulmonary arteries. Atypical locations include abdominal veins (splanchnic, renal, and ovarian), cerebral veins and venous dural sinuses, and the upper extremity venous system and retinal vein thrombosis. USVT are characterized by distinct pathophysiological and clinical features, and their incidence is heterogeneous (Figure 1).

In the absence of absolute contraindications, the anticoagulation therapy should be started after USVT diagnosis.

Direct oral anticoagulants (DOACs) are the preferred treatment for most patients with deep vein thrombosis (DVT) of the lower extremities and/or pulmonary embolism (PE) [1,2,3]. These drugs provide several advantages over vitamin K antagonists (VKAs). These include a predictable anticoagulant effect, fewer drug interactions, no food interactions, and a rapid onset of action, allowing for fixed-dose administration. Additionally, DOACs are associated with a lower risk of major bleeding events, particularly intracranial hemorrhages (ICHs) [4,5,6]. These benefits make them potentially suitable anticoagulants for patients with USVT.

In contrast to these advantages, there is a limited availability of reversal agents for DOACs in cases of life-threatening bleeding or emergency surgery. Alongside the need for more reversal agents, there is uncertainty regarding the plasma concentrations of DOACs. While routine laboratory monitoring is not required, measuring drug levels may be necessary in several circumstances, such as in patients with extreme body weight, impaired renal function, emergency surgical procedures, or major bleeding events.

Because the pivotal phase 3 randomized clinical trials (RCTs) conducted on DOACs did not include patients with USVT [7], overall, the international guidelines on thrombosis recommend parenteral anticoagulation with unfractionated heparin (UFH) or low-molecular-weight heparin (LMWH), followed by VKA anticoagulant therapy for these patients [8]. However, in recent years, evidence has emerged from a few RCTs and large observational studies assessing the safety and effectiveness of DOACs in patients with USVT.

This narrative review aims to analyze the latest evidence on the use of DOACs in patients with USVT, including cerebral, splanchnic, upper extremity, ovarian, renal, and retinal vein thrombosis. It also seeks to provide practical recommendations for their use in this clinical setting based on the most up-to-date scientific literature.

## 2. Methods

### 2.1. Literature Search

Using PubMed, Web of Science, Scopus, EMBASE, Cochrane Library, and the Cochrane Central Register of Controlled and Trials, we searched for the literature on DOACs for the treatment of patients with USVT from January 2010 to December 2024. The following retrieval strategy was employed: “Unusual-site venous thrombosis (MeSH word or text word), Direct oral anticoagulants (MeSH word or text word), Cerebral venous thrombosis (MeSH word or text word), Ovarian vein thrombosis (MeSH word or text word), Renal vein thrombosis (MeSH word or text word), Retinal vein occlusion (MeSH word or text word), Upper extremity deep vein thrombosis (MeSH word or text word), Splanchnic vein thrombosis (MeSH word or text word), Randomized Clinical Trial (MeSH word or text word) and Observational study (MeSH word or text word)”. The search strategy applied in each database was composed of a combination of these terms in the heading. The search was limited to studies on humans without language restrictions.

### 2.2. Study Selection

The article’s inclusion criteria were as follows: (1) hospitalized or outpatients; (2) the study analyzed the efficacy and safety of DOACs in patients with USVT. The exclusion criteria were as follows: (1) review articles; (2) duplicate publications; (3) studies without usable data. Cross-sectional studies, case series, case reports, and conference abstracts were included if relevant data were provided. To aid the screening process, the investigators used a standardized screening form.

### 2.3. Data Extraction

Using a standardized form, data from the included studies were extracted and reviewed by the investigators. The following data were extracted: (1) study setting (country, year of publication, and data collection period), (2) sample size, (3) DOAC, (4) USVT type, (5) period of anticoagulation (months), (6) median follow-up (days), (7) recurrent thrombosis, (8) intracranial hemorrhage, and (9) major extracranial hemorrhage.

## 3. Evidence Regarding DOACs in USVT

### 3.1. Cerebral Venous and Dural Sinus Thrombosis

Cerebral vein thrombosis (CVT) encompasses thrombosis of the dural venous sinuses and the cortical or deep cerebral veins. CVT accounts for 0.5% to 3% of all strokes. The pathogenesis of CVT is multifactorial, mainly due to the anatomical variability of the venous system. CVT, or dural sinus thrombosis, obstructs venous drainage, leading to increased venous and capillary pressure. This elevation in pressure disrupts the blood–brain barrier and raises intracranial pressure due to reduced cerebrospinal fluid absorption. The consequent increase in pressure can induce vasogenic edema, trigger venous hemorrhage, and reduce cerebral blood flow. Several underlying risk factors may contribute to CVT. The main risk factors include the use of estrogen-containing oral contraceptives and pregnancy or the puerperium [9]. Additional risk factors include central nervous system infections, thrombophilia, and malignancies [10]. Finally, no identifiable risk factor can be identified in a small subset of cases. The clinical presentation of CVT is diverse, including headaches and seizures, and requires a high level of clinical suspicion. Its diagnosis is based on magnetic resonance imaging or computed tomography.

Guidelines recommend anticoagulant therapy for patients with CVT [11,12]. The objectives of anticoagulants are to promote their resolution and prevent recurrent events. Notably, anticoagulant treatment is safe even in patients with CVT who develop intracranial hemorrhages. During the acute phase of CVT, the preferred anticoagulant is heparin (LMWH over UFH). For long-term management, guidelines recommend VKAs while maintaining the international normalized ratio (INR) within the standard therapeutic range of 2.0 to 3.0 [11].

Emerging evidence suggests that DOACs may also be a reasonable choice for oral anticoagulation in selected patients with CVT. ACTION-CVT, a large multicenter international retrospective study, included 845 consecutive patients with CVT treated with warfarin (52%) or DOACs (33%) versus both at different times (15%) from January 2015 to December 2020 [13]. Among patients who used DOACs, 13.5% used dabigatran, 18.2% used rivaroxaban, 66.6% used apixaban, and 1.7% used other or multiple DOACs. The mean age was 44.8 years, and 64.7% were women. The median follow-up was 345 days. When compared with warfarin, DOAC treatment was associated with a similar risk of recurrent venous thrombosis (Hazard Ratio [HR], 0.94; 95% Confidence Interval [CI], 0.51–1.73; *p* = 0.84), death (HR, 0.78; 95% CI, 0.22–2.76; *p* = 0.70), and rate of partial/complete recanalization (HR, 0.92; 95% CI, 0.48–1.73; *p* = 0.79), but a lower risk of major bleeding events (HR, 0.35; 95% CI, 0.15–0.82; *p* = 0.02), especially intracranial bleeding (*p* = 0.04). Another cohort study compared events in 111 consecutive individuals with CVT who were prescribed VKAs (66) versus DOACs (45) as part of their routine clinical care [14]. The DOACs used included rivaroxaban (36) and dabigatran (9). There was no recurrent thrombotic event in either group, with two major hemorrhages in the DOACs group and four (one intracerebral hemorrhage) in the warfarin group. There were no differences in recanalization rates (four in either arm). In a recently published study, CVT databases from seven academic medical centers that included patients with CVT were retrospectively analyzed [15]. Four hundred and three patients were included. Of them, 48 (12%) were treated with apixaban, and 355 (88%) received VKAs. There were no differences in efficacy or safety outcomes, including the complete recanalization rates, intracranial hemorrhages, and CVT recurrences between groups. The study conducted by Khorvash et al. was a masked case–control study that included 50 CVT patients allocated 1:1 to rivaroxaban 20 mg daily or warfarin (target INR, 2.0–3.0) for 3 months [16]. The primary outcomes assessed were the Modified Rankin Scale (mRS) scores and the major bleeding rate at discharge and 3- and 6-month follow-ups. No significant differences were found between the rivaroxaban and warfarin groups at any point during the follow-up. Both groups improved their mRS scores, and no major hemorrhagic events were reported. Another retrospective study did not find significant differences in clinical outcomes among patients using different types of anticoagulation therapy (DOACs/AVKs/LMWH). However, the size of the simple DOACs that were received was limited [17].

Five RCTs have analyzed the efficacy and safety of DOACs in patients with CVT (Table 1). Of these, two trials investigated dabigatran [18,19], and three investigated rivaroxaban [20,21,22]. RE-SPECT CVT was an international prospective clinical trial that randomized 120 individuals with CVT 1:1 to warfarin with a target INR of 2.0 to 3.0 or dabigatran of 150 mg twice daily for 6 months after 5 to 15 days of lead-in parenteral anticoagulation with LMWH or UFH [18]. There was no recurrent CVT in either group, with one (1.7% [95% CI, 0.0–8.9]) major hemorrhage (gastrointestinal bleeding) in the dabigatran group and two (3.3% [95% CI, 0.4–11.5], both intracerebral hemorrhages) in the warfarin group. In a post hoc analysis, complete recanalization occurred in 44% and 36%, and partial recanalization in 42% and 49% in the dabigatran and AVK groups, respectively, after 24 weeks of treatment, respectively (*p* = 0.44) [23]. In the CHOICE-CVT trial, 89 patients were enrolled to receive either dabigatran (44 patients) or warfarin (45 patients) [19]. By day 180, the dabigatran group had a higher, though not statistically significant, incidence of recurrent thrombosis (18.2% versus 6.7%; *p* = 0.099) compared to the warfarin group. Both groups had no cases of major bleeding and similar rates of non-major bleeding. SECRET was a phase 2 trial conducted at 12 Canadian centers that randomized 55 participants with CVT 1:1 to rivaroxaban of 20 mg daily versus warfarin (target INR, 2.0 to 3.0) or LMWH for a minimum of 6 months [20]. The primary outcome was a composite of symptomatic intracranial hemorrhages, major extracranial hemorrhages, or mortality. At 6 months, there was one recurrent CVT (3.8% [95% CI, 0.1–19.6]), one intracerebral hemorrhage (3.8% [95% CI, 0.1–19.6]), and two clinically relevant non-major extracranial bleeding events (7.7% [95% CI, 0.9–25.1]) in the rivaroxaban group with no recurrence or bleeding events in the control group. In the study by Maqsood et al., 45 adult CVT patients were randomized to receive rivaroxaban of 20–30 mg daily (n = 21) or warfarin (n = 24) for 3–12 months after 5 to 12 days of lead-in parenteral anticoagulation [21]. No major bleeding events or thrombosis recurrence were reported in either group. By 6 months, overall recanalization was achieved in 86% (18) of patients in the DOAC group and 83% (20) in the warfarin group. In the substudy of the EINSTEIN-Jr trial, after initial heparinization, 114 children with CVT were randomized in a 2:1 ratio to receive treatment either with rivaroxaban or standard anticoagulants (continuing on heparin or switching to VKAs) [22]. None of the 73 children treated with rivaroxaban and 1 (2.4%) of the 41 treated with standard anticoagulants suffered symptomatic recurrent thrombosis. Clinically relevant bleeding occurred in five (6.8%; all non-major and non-cerebral) of the rivaroxaban-treated children and in one (2.5%; major [subdural] bleeding) of the standard anticoagulant-treated children (absolute difference, 4.4%; 95% CI, −6.7% to 13.4%).

A recent systematic review encompassing the three randomized trials and 16 observational cohort studies comparing DOACs and VKAs for the treatment of CVT showed comparable outcomes between the two groups in terms of recurrent CVT, major bleeding, and complete recanalization rates (42.9% and 42.3%, respectively; HR, 0.98 [95% CI, 0.87–1.11]) [24].

The ongoing prospective DOAC-CVT study will contribute to the growing body of evidence on this topic (ClinicalTrials.gov NCT04660747) [25]. Additionally, two clinical trials are being conducted to evaluate: rivaroxaban versus apixaban (ClinicalTrials.gov NCT06531122), and dabigatran versus apixaban (ClinicalTrials.gov NCT06551415).

### 3.2. Ovarian Vein Thrombosis

Ovarian vein thrombosis (OVT) typically occurs postpartum, usually presenting within 4 weeks of delivery. Other risk factors for OVT include the use of estrogen-containing oral contraceptives, gynecological malignancies, abdominopelvic surgery, pelvic inflammatory disease, and septic pelvic thrombophlebitis. In 4% to 16% of cases, OVT is classified as unprovoked [26].

The 2012 guidelines of the British Society for Haematology on the management of USVT recommends conventional anticoagulation treatment for 3–6 months for postpartum OVT while incidentally identifying OVT in patients who have undergone hysterectomy and salpingo-oophorectomy, and OVT cancer-associated isolated anticoagulation is overall not advised except for patients with inferior vena cava extension or pulmonary embolism [26].

To date, no RCTs have evaluated the use of DOACs in women with OVT. The available evidence is derived from observational studies and case reports. In the study by Janczak et al., eight patients with OVT treated with DOACs were included in a broader analysis of venous thrombosis in atypical locations [27]. Lenz et al. reported that 11 of 219 women with OVT received direct factor Xa inhibitors [28]. Additionally, five case reports documented the use of rivaroxaban [29,30,31,32,33], and three evaluated apixaban [34,35,36]. All cases achieved favorable outcomes. Moreover, Covut et al. retrospectively analyzed 36 women with OVT treated between 2012 and 2018 [37]. Of these, 10 received DOACs (rivaroxaban or apixaban), 11 were treated with VKAs, and 15 with LMWH. Complete recanalization during follow-up was achieved in 70% of the DOAC group, compared to 55% in the VKAs group and 93% in the LMWH group. The 1-year cumulative incidence of clinically relevant bleeding was 10%, 18%, and 25%, respectively. Although these findings are encouraging, they should be interpreted cautiously, given the lack of robust evidence.

### 3.3. Renal Vein Thrombosis

There are scarce data on the prevalence of renal vein thrombosis (RVT) in adults, as it is often asymptomatic and resolves spontaneously. Nephrotic syndrome is the most common risk factor for RVT [38]. Other risk factors include malignancies (particularly renal cell carcinoma), abdominal surgery, trauma, infections, and post-renal transplant complications [39,40,41]. Approximately 12% of RVT cases are classified as unprovoked [39,40]. Additionally, nearly two-thirds of patients present with bilateral renal vein involvement [38].

There is limited evidence regarding treating RVT. Acute RVT is typically managed with parenteral anticoagulation (UFH or LMWH), followed by VKAs [26]. Recent studies suggest an anticoagulant treatment duration of 3–12 months for patients with RVT caused by a transient risk factor and a longer duration for those with permanent hypercoagulable states or unprovoked RVT [38,41].

No RCTs have been conducted on the use of DOACs in patients with RVT. The previously mentioned Janczak et al. study included three RVT patients treated with DOACs [27]. Furthermore, several published case reports have examined the use of rivaroxaban [42,43,44], apixaban [45,46,47], and edoxaban [48] in RVT patients, reporting favorable clinical outcomes.

### 3.4. Retinal Vein Occlusion

The classification of retinal vein occlusion (RVO) is based on the anatomical location of the occlusion: branch retinal vein occlusion and central retinal vein occlusion. The pathophysiology of both conditions differs. Branch retinal vein occlusion is believed to occur due to the blockage of retinal veins at arteriovenous crossings. It has been hypothesized that an inflexible atherosclerotic arteriole compresses the more distensible veins, causing venous occlusion. In the case of central retinal vein occlusion, pathophysiology primarily involves the formation of a thrombus consisting of fibrin and platelets in the vessel.

The current treatment for RVO, as recommended by the Royal College of Ophthalmologists [49] and the European Society of Retina Specialists (EURETINA) [50] guidelines, includes the use of intravitreal antivascular endothelial growth factor (anti-VEGF) agents, laser photocoagulation, and corticosteroids. Anticoagulant treatment is an additional option but is not routinely recommended due to the lack of robust evidence. However, anticoagulant therapy may be considered in patients without local risk factors (e.g., glaucoma), with recent symptom onset, or with major thrombosis risk factors such as antiphospholipid antibodies [49,50,51]. In these cases, LMWH at therapeutic doses for 10 to 15 days, followed by half-therapeutic doses for up to 3 months, is suggested [51].

There are no RCTs evaluating the use of DOACs for the treatment of RVO. Three published case reports have described the use of DOACs in patients with RVO, all reporting favorable outcomes. Two cases involved apixaban following mRNA SARS-CoV-2 vaccination [52,53], and one involved rivaroxaban in a patient with severe COVID-19 pneumonia and bilateral central RVO [54].

### 3.5. Upper Extremity Deep Vein Thrombosis

Upper extremity deep vein thrombosis (UEDVT) accounts for approximately 5% to 10% of all DVT cases [55]. UEDVT involves thrombosis in the axillary, subclavian, internal jugular, or brachiocephalic veins (proximal veins) and the brachial, radial, or ulnar veins (distal veins). UEDVT is classified as either primary (10% of cases) or secondary (90% of cases). Primary UEDVT includes effort-related thrombosis (Paget–von Schrötter syndrome), a thrombotic manifestation of thoracic outlet syndrome, and unprovoked thrombotic events [56,57]. Secondary UEDVT is typically associated with indwelling devices, such as central venous catheters, pacemaker or defibrillator leads, tunneled central access lines, or malignancy [56].

The cornerstone of treatment for UEDVT is anticoagulation. For primary UEDVT, the American College of Chest Physicians (ACCP) guidelines recommend anticoagulant therapy over thrombolysis [58]. However, thrombolysis may be considered in patients with severe symptoms, extensive thrombus involving the subclavian and axillary veins, symptom onset within 14 days, good performance status, and a low risk of bleeding [58]. In cases of thoracic outlet syndrome, surgical decompression is required. This procedure involves the first rib resection, the costoclavicular ligament removal, and anterior scalenectomy [59]. For most patients, anticoagulant therapy is suggested for 3–6 months, except for those with active cancer. In catheter-related (CVC associated) UEDVT, catheter removal is unnecessary if the device is well positioned, functional, uninfected, and required. However, anticoagulation should continue for as long as the catheter remains [60].

To date, one RCT has evaluated the use of DOACs for treating UEDVT (Table 1). In the ARM-DVT trial (ClinicalTrials.gov NCT02945280), 52 patients with acute UEDVT received apixaban of 10 mg twice daily for 7 days, followed by apixaban of 5 mg twice daily for 11 weeks. The primary efficacy was the composite of a ninety-day rate of new or recurrent objectively confirmed symptomatic venous thrombosis and thrombosis-related death. The primary safety outcome was the composite of major and clinically relevant non-major bleeding. The study concluded recruitment in November 2021; however, the results are not yet provided [61].

Several large cohort studies have been published in recent years. Houghton et al. prospectively assessed a cohort of 210 patients with acute UEDVT treated at the Mayo Clinic between 2013 and 2019 [62]. One hundred and two patients were treated with DOACs (63 with apixaban and 39 with rivaroxaban) and 108 with LMWH/VKAs. At the 3-month follow-up, outcomes were similar between the two groups regarding venous thromboembolism (VTE) recurrence (1.0% versus 0.9%), major bleeding (0% versus 2.8%), and all-cause mortality (2.0% versus 4.6%). Montiel et al. described a cohort of 55 UEDVT patients from the Swedish national anticoagulation registry who received rivaroxaban (46), apixaban (7), and dabigatran (2). Low outcome rates were reported at the 6-month follow-up, including thrombosis recurrence (2%), major bleeding (0%), and mortality (0%) [63]. An Italian multicenter retrospective study enrolled 61 patients with non-cancer, non-CVC-related UEDVT treated with DOACs, including rivaroxaban (60.7%), apixaban (18.0%), dabigatran (11.5%), and edoxaban (9.8%) [64]. While DOACs were often preceded by LMWH or fondaparinux, a single-drug approach was used in some cases. Most patients (two-thirds) were treated for 3–6 months, with the remainder receiving >6 months of therapy. All patients showed partial or complete recanalization, with no episodes of VTE recurrence, pulmonary embolism, major bleeding, or mortality. In the Canadian prospective multicenter CATHETER-2 study, 70 cancer patients with CVC-related UEDVT were treated with rivaroxaban, with LMWH used initially in 51% of cases [65]. While only one thrombosis recurrence (a fatal PE confirmed by autopsy) was reported, the study raised concerns about bleeding risks, including seven major bleeding events and four clinically relevant non-major bleeding episodes during the 3-month follow-up. Another prospective study analyzed 188 patients with acute UEDVT, 33% of whom had catheter- or pacemaker-related thrombosis [66]. DOAC treatment lasted an average of 5.1 ± 2.8 months, with event rates per 100 patient-years of recurrent VTE (0.9), major bleeding (1.7), and all-cause mortality (6.0). No fatal bleeding or fatal VTE recurrence was observed. Finally, in a recently published retrospective study conducted in Australia between December 2010 and December 2022, 137 patients with UEDVT were analyzed [67]. In the long-term anticoagulation treatment, 48 patients received LMWH, 30 were treated with VKAs, and 53 with DOACs. The analysis revealed no differences in the rates of major bleeding and clot progression/recurrence among the different cohorts.

An ongoing single-arm study is evaluating the efficacy and safety of apixaban for this indication (ClinicalTrials.gov NCT03100071).

### 3.6. Splanchnic Vein Thrombosis

Splanchnic vein thrombosis (SVT) has an incidence rate ranging from 0.7 to 2.7 cases per 100,000 person-years in the general population [68]. SVT includes portal vein thrombosis (PVT), mesenteric venous thrombosis (MVT), splenic vein thrombosis, and Budd–Chiari syndrome. Depending on the location of the thrombosis, patients may be at risk of developing liver insufficiency, portal hypertension, or bowel infarction.

MVT is a multifactorial disorder influenced by both inherited and acquired risk factors. Local intra-abdominal inflammations, such as pancreatitis and inflammatory bowel disease, appear to be associated with thrombus formation in the large veins. In contrast, systemic hypercoagulable states lead to thrombosis in the intramural venules. MVT typically affects the distal small intestine and rarely the colon, with acute occlusion leading to decreased perfusion, venous congestion, and subsequent bowel wall edema, which can result in bowel infarction and ischemia. PVT in patients with healthy livers is often attributed to hereditary prothrombotic states. Hereditary conditions associated with PVT include Factor V Leiden mutation, prothrombin gene mutation, protein C and S deficiencies, and antithrombin deficiency. In patients with cirrhosis, pathogenesis likely involves imbalanced hemostasis and slowed portal flow. Additionally, acquired conditions linked to PVT include hepatocellular carcinoma, chronic myeloproliferative disorders, antiphospholipid syndrome, paroxysmal nocturnal hemoglobinuria, Behçet’s syndrome, and abdominal inflammatory lesions.

The primary treatment for splanchnic vein thrombosis (SVT) is anticoagulation therapy, which should be initiated promptly to enhance the likelihood of vessel recanalization and minimize the risk of complications associated with portal hypertension. Guidelines from the American Association for the Study of Liver Diseases (AASLD) [69] and the European Association for the Study of the Liver (EASL) [70] recommend a six-month course of anticoagulation for acute SVT, with the possibility of extended treatment depending on the underlying cause. The choice of anticoagulant drug can be challenging. The AASLD recommends an individualized approach, including LMWH, VKAs, and DOACs, with the caution that data on the efficacy and safety of DOACs in this patient group are limited [64]. Expert opinion suggests that for patients listed for liver transplantation, VKAs should be used due to their widespread availability of reversal agents and the need for surgery on short notice [68].

Three clinical trials have evaluated the role of DOACs in SVT (Table 1). A randomized trial of rivaroxaban versus no anticoagulation in patients with non-cirrhotic PVT was terminated at a median of 11.8 months after observing thrombosis recurrence rates of 0 versus 19.7 per 100 person-years, respectively [71]. The RIVA-SVT100 study evaluated the safety and efficacy of rivaroxaban in 100 non-cirrhotic patients with acute SVT [72]. Patients received 15 mg twice daily for 3 weeks, followed by 20 mg once daily for 3 months. The study reported over 80% recanalization, with 47% achieving complete recanalization and low rates of recurrent thrombosis (2.1%) and major bleeding (2.1%). A recent meta-analysis suggested either noninferiority or superiority of DOACs compared with VKAs and LMWH [73].

Patients with cirrhosis with PVT merit special consideration for anticoagulation as disturbances in liver function and hemostatic–thrombotic balance may affect the safety and efficacy of anticoagulation. A meta-analysis of 11 studies of patients with cirrhosis confirmed higher portal thrombosis recanalization in the DOAC group compared to AVKs (87% versus 44%, respectively), and no difference in variceal bleeding or death was observed between groups [74]. A more recent systematic review evaluated the comparative effectiveness and safety of DOACs versus AVKs/LMWH in cirrhotic patients with PVT [75]. The review included four observational studies from Japan, China, and the USA, covering 223 patients. Results showed that DOACs, particularly edoxaban, effectively improved recanalization rates and reduced PVT progression in cirrhotic patients without significantly increasing bleeding risks. The last update of the American Gastroenterological Association (AGA) guidelines suggests the use of DOACs in PVT patients with Child–Turcotte–Pugh (CTP) class A and with caution in CTP class B but are not advised in CTP class C [76].

**Table 1 pharmaceutics-17-00342-t001:** Clinical trials with direct oral anticoagulants for the treatment of unusual-site venous thrombosis.

	RE-SPECT CVT [18]	CHOICE-CVT [19]	SECRET [20]	Maqsood et al. [21]	EINSTEIN-Jr CVT [22]	ARM-DVT [61]	RIPORT [71]	RIVA-SVT100 [72]
USVT	CVT	CVT	CVT	CVT	CVT	UEDVT	Non-cirrhotic chronic PVT	Non-cirrhotic SVT (portal, mesenteric, and splenic veins)
Study period	21 December 2016 to 22 June 2018	October 2017 to February 2023	March 2019 to October 2021	May 2017 to May 2018	November 2014 to the first quarter of 2019	NA	September 2015 to January 2020	June 2015 to March 2021
Study setting	France, Germany, India, Italy, the Netherlands, Poland, Portugal, Russia, and Spain	China	Canada	Pakistan	Australia, Turkey, Israel, China, countries in Europe, South America, and North America	United States	France	Italy, Canada, France, and Germany
DOAC	Dabigatran, 150 mg twice daily	Dabigatran, 150 mg twice daily	Rivaroxaban, 20 mg daily	Rivaroxaban, 20–30 mg daily	Rivaroxaban (bodyweight-adjusted dose), after initial heparinisation	Apixaban, 10 mg twice daily for 7 days, followed by apixaban, 5 mg twice daily	Rivaroxaban, 15 mg daily	Rivaroxaban, 15 mg twice daily for 3 weeks, followed by 20 mg once daily
Comparison	Warfarin (target INR, 2.0–3.0)	Warfarin (target INR, 2.0–3.0)	Warfarin (target INR, 2.0–3.0) or LMWH	Warfarin (target INR, 2.0–3.0)	VKA (target INR, 2.0–3.0), after initial heparinisation	Warfarin (target INR, 2.0–3.0) or LMWH	No anticoagulation	None (single-arm study)
Phase	2	NA	2	NA	3	3	2	Pilot/interventional
Minimum period of anticoagulation, months	6	6	6	3	3	3	6	3
Sample size	120	89	55	45	114	357	111	100
Median follow-up, days	175	180	180	365	90	90	909	180
Recurrent thrombosis (DOAC/control group), n	0/0	8/3	1/0	0/0	0/1	NA	0/10	2
Intracranial hemorrhage (DOAC/control group), n	0/2	NA	1/0	0/0	0/1	NA	0/0	0
Major extracranial hemorrhage (DOAC/control group), n	1/0	0/0	2/0	0/0	0/0	NA	2/1	3
Conclusions	Dabigatran demonstrated non-inferiority compared to warfarin	The findings support the consideration of dabigatran etexilate therapy	Rivaroxaban demonstrated non-inferiority compared to warfarin/LMWH	Rivaroxaban demonstrated non-inferiority compared to warfarin	Rivaroxaban had a low risk of recurrent thrombosis and major bleeding events	NA	Rivaroxaban significantly reduced the incidence of recurrent venous thrombosis without increasing major bleeding events	Rivaroxaban demonstrated efficacy and safety for the treatment of non-cirrhotic SVT

Abbreviations: CVT, Cerebral venous thrombosis; DOAC, direct oral anticoagulant; LMWH, low-molecular-weight heparin; NA, not available; SVT, splanchnic vein thrombosis. UEDVT, upper extremity deep vein thrombosis; USVT, unusual-site venous thrombosis; VKA, vitamin K antagonist.

## 4. Conclusions

In the latest update, the British Society for Haematology (BSH) guidelines on the management of USVT recommend considering DOACs as an alternative to VKAs for patients with CVT and UEDVT, and for the treatment of non-neoplastic SVT [77] (Table 2). However, no specific recommendations are provided for patients with Budd–Chiari syndrome or those with cirrhosis. The International Society on Thrombosis and Haemostasis (ISTH) guidelines only address SVT [68] (Table 2). For non-cirrhotic and cancer patients with SVT, guidelines suggest DOACs over LMWH and VKAs. In cirrhotic patients, LMWH is recommended. For cancer-associated SVT, LMWH or DOACs are recommended. However, LMWH is preferred in patients with luminal gastrointestinal cancer, active gastrointestinal mucosal abnormalities, genitourinary cancer at a high risk of bleeding, or those receiving systemic therapies with significant drug–drug interactions with DOACs. Finally, the guidance document of the Baveno VII consensus has included the possibility of using DOACs for the treatment of SVT [78].

Evidence on the use of DOACs for treating USVT is limited and primarily derived from a few RCTs and several observational cohort studies. Conducting RCTs in USVT patients is challenging due to the low incidence rate. However, the available evidence suggests that DOACs can be used in selected USVT patients, as they demonstrate comparable effectiveness and a potential trend toward better safety compared to VKAs. Caution is advised in patients with special conditions, such as those with liver cirrhosis, malignancy, advanced renal failure, severe liver dysfunction, or high-risk antiphospholipid syndrome (mainly patients with prior arterial thrombosis and triple positivity for antiphospholipid antibodies).

Our review highlights critical gaps in the use of DOACs in USVT treatment that need addressing through further research. While DOACs have shown promising results in USVT patients, it is essential to emphasize the need for long-term studies to provide more robust data on efficacy and safety over time. This is not only to extend the duration of studies but also to delve into specific scenarios such as PVT related to hepatocellular carcinoma or catheter-related upper-extremity thrombosis in cancer patients. Ongoing RCT data will be crucial in developing tailored therapeutic strategies and guidelines to enhance patient outcomes in these complex conditions.

## Figures and Tables

**Figure 1 pharmaceutics-17-00342-f001:**
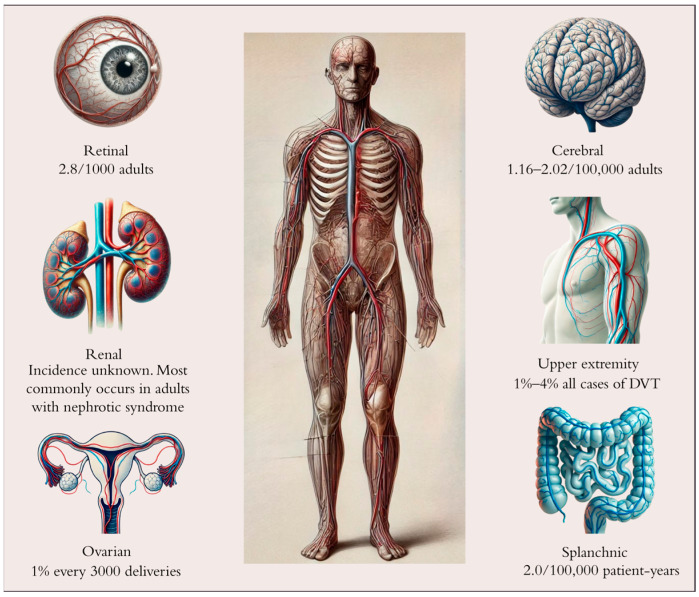
Estimated incidence of unusual-site venous thrombosis.

**Table 2 pharmaceutics-17-00342-t002:** Recommendations of the guidelines regarding the use of direct oral anticoagulants in patients with unusual-site venous thrombosis.

Guidelines	USVT	Recommendation
British Society for Haematology [77]	CVT	Dabigatran can be used, after initial treatment with heparins (1B). Direct factor Xa inhibitors can be considered, after initial treatment with heparins (2C).
	UEDVT	DOACs can be considered as an alternative to VKAs (2C).
	SVT *^,†^	DOACs can be considered as an alternative to VKAs (2C).
International Society on Thrombosis and Haemostasis [68]	CVT	Not provided.
	UEDVT	Not provided.
	SVT non-associated with cirrhosis or cancer	DOACs are recommended over LMWH and VKAs.
	SVT associated with cirrhosis	LMWH with switch to DOACs or VKAs are recommended if the severity of liver dysfunction does not contraindicate them.
	SVT associated with cancer	LMWH or DOACs are recommended. LMWH is preferred in patients with luminal gastrointestinal cancer, active gastrointestinal mucosal abnormalities, genitourinary cancer at high risk of bleeding, or receiving current systemic therapy with potentially relevant drug–drug interactions with DOACs.
Baveno VII [78]	SVT non-associated with cirrhosis	As a primary treatment option for recent PVT, start with LMWH and switch to VKAs (B1). DOACs can be considered the primary option in selected cases in the absence of triple positive anti-phospholipid syndrome, although data are limited (C2).
	SVT associated with cirrhosis	DOACs can be considered in patients with Child–Pugh class A cirrhosis. DOACs should be used with caution in patients with Child–Pugh class B cirrhosis. The use of DOACs in those with Child-Pugh class C cirrhosis is not recommended (C2).

Abbreviations: VKAs, vitamin K antagonists; CVT, cerebral venous thrombosis; DOACs, direct oral anticoagulants; LMWH, low-molecular-weight heparin; SVT, splanchnic vein thrombosis. UEDVT, Upper extremity deep vein thrombosis; USVT, unusual-site venous thrombosis. * include portal vein thrombosis, hepatic vein thrombosis, mesenteric vein thrombosis, and splenic vein thrombosis. ^†^ non-associated with cirrhosis or cancer.

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
