# Peer review of "Direct Oral Anticoagulants for the Treatment of Unusual-Site Venous Thrombosis: An Update"

_pharmaceutics, 2025, doi:10.3390/pharmaceutics17030342_

Round 1
Reviewer 1 Report
Comments and Suggestions for Authors
This review provides a valuable update on the expanding role of DOACs in managing USVT. However, several areas need improvement to enhance the clarity, depth, and overall quality of the article.
While the review indicates that it examines evidence from randomized controlled trials (RCTs) and large observational studies, it would benefit from a more detailed explanation of the criteria for study inclusion. A clear description of the search strategy, inclusion/exclusion criteria, and the data extraction process would enhance the rigor and transparency of the review.
The review heavily emphasizes the clinical application of DOACs but would be strengthened by a deeper exploration of the underlying pathophysiology of venous thrombosis at unusual sites. Understanding the unique challenges associated with thrombosis in areas such as the cerebral, splanchnic, or retinal veins is essential for clinicians. A more thorough discussion of these mechanisms could help contextualize the use of DOACs in such conditions.
Moreover, the review lacks a detailed discussion on the long-term outcomes of patients treated with DOACs for USVT. Information regarding the long-term efficacy and safety of these drugs in unusual sites, particularly in relation to recurrent thrombosis or bleeding complications, would be invaluable for clinicians considering DOAC therapy in these contexts.
The conclusion should highlight areas where further research is needed. For instance, the review could emphasize gaps in the literature, such as the scarcity of RCTs specifically focused on USVT or the need for more data on the long-term safety of DOACs in these patient populations. Future research directions could explore alternative treatment options, personalized therapy strategies, or the development of specific guidelines for DOAC use in uncommon venous thrombotic conditions.
Author Response
Reviewer 1 This review provides a valuable update on the expanding role of DOACs in managing USVT. However, several areas need improvement to enhance the clarity, depth, and overall quality of the article. While the review indicates that it examines evidence from randomized controlled trials (RCTs) and large observational studies, it would benefit from a more detailed explanation of the criteria for study inclusion. A clear description of the search strategy, inclusion/exclusion criteria, and the data extraction process would enhance the rigor and transparency of the review.
We have added a new section: Methods. In this section we detail the search strategy, inclusion/exclusion criteria, and the data extraction process. “2. Methods 2.1 Literature Search Using PubMed, Web of Science, Scopus, EMBASE, Cochrane Library, Cochrane Central Register of Controlled and Trials, we searched for literature on DOACs for the treatment of patients with USVT from January 2010 to December 2024. The following retrieval strategy was employed: “Unusual-site venous thrombosis (MeSH word or text word), Direct oral anticoagulants (MeSH word or text word), Cerebral venous thrombosis (MeSH word or text word), Ovarian vein thrombosis (MeSH word or text word), Renal vein thrombosis (MeSH word or text word), Retinal vein occlusion (MeSH word or text word), Upper extremity deep vein thrombosis (MeSH word or text word), Splanchnic vein thrombosis (MeSH word or text word), Randomized Clinical Trial (MeSH word or text word) and Observational study (MeSH word or text word). The search strategy applied in each database was composed of a combination of these terms in the heading. The search was limited to studies on humans without language restrictions. 2.2 Study selection The article's inclusion criteria were as follows: (1) hospitalized or outpatients; (2) the study analyzed the efficacy and safety of DOACs in patients with USVT. The exclusion criteria were as follows: (1) review articles; (2) duplicate publications; and (3) studies without usable data. Cross-sectional studies, case series, case reports, and conference abstracts were included if relevant data was provided. To aid the screening process, the investigators used a standardized screening form. 2.3 Data extraction Using a standardized form, data from the included studies were extracted and reviewed by the investigators. The following data were extracted: (1) study setting (country, year of publication, data collection period), (2) sample size, (3) DOAC, (4) USVT type, (5) period of anticoagulation (months), (6) median follow-up (days), (7) recurrent thrombosis, (8) intracranial haemorrhage and (9) major extracranial haemorrhage”.
The review heavily emphasizes the clinical application of DOACs but would be strengthened by a deeper exploration of the underlying pathophysiology of venous thrombosis at unusual sites. Understanding the unique challenges associated with thrombosis in areas such as the cerebral, splanchnic, or retinal veins is essential for clinicians. A more thorough discussion of these mechanisms could help contextualize the use of DOACs in such conditions.
Thank you for your comment. We have incorporated the pathophysiological mechanisms underlying in: -CVT: “The pathogenesis of cerebral venous thrombosis (CVT) is multifactorial, mainly due to the anatomical variability of the venous system. CVT, or dural sinus thrombosis, obstructs venous drainage, leading to increased venous and capillary pressure. This elevation in pressure disrupts the blood-brain barrier and raises intracranial pressure due to reduced cerebrospinal fluid (CSF) absorption. The consequent increase in pressure can induce vasogenic edema, trigger venous hemorrhage, and reduce cerebral blood flow” -SVT: “MVT is a multifactorial disorder influenced by both inherited and acquired risk factors. Local intra-abdominal inflammations, such as pancreatitis and inflammatory bowel disease, appear to be associated with thrombus formation in the large veins. In contrast, systemic hypercoagulable states lead to thrombosis in the intramural venules. MVT typically affects the distal small intestine and rarely the colon, with acute occlusion leading to decreased perfusion, venous congestion, and subsequent bowel wall edema, which can result in bowel infarction and ischemia. PVT in patients with healthy livers is often attributed to hereditary prothrombotic states. Hereditary conditions associated with PVT include Factor V Leiden mutation, prothrombin gene mutation, protein C and S deficiencies and antithrombin deficiency. In patients with cirrhosis, pathogenesis likely involves imbalanced hemostasis and slowed portal flow. Additionally, acquired conditions linked to PVT include hepatocellular carcinoma, chronic myeloproliferative disorders, antiphospholipid syndrome, paroxysmal nocturnal hemoglobinuria, Behçet's syndrome, and abdominal inflammatory lesions”. -RVO: “The classification of retinal vein occlusion (RVO) is based on the anatomical location of the occlusion: branch retinal vein occlusion and central retinal vein occlusion. The pathophysiology of both conditions differs. Branch retinal vein occlusion is believed to occur due to the blockage of retinal veins at arteriovenous crossings. It has been hypothesized that an inflexible atherosclerotic arteriole compresses the more distensible veins, causing venous occlusion. In the case of central retinal vein occlusion, pathophysiology primarily involves the formation of a thrombus consisting of fibrin and platelets in the vessel”.
Moreover, the review lacks detailed discussion on the long-term outcomes of patients treated with DOACs for USVT. Information regarding the long-term efficacy and safety of these drugs in unusual sites, particularly in relation to recurrent thrombosis or bleeding complications, would be invaluable for clinicians considering DOAC therapy in these contexts.
Thank you for your comments regarding the long-term outcomes in patients treated with direct oral anticoagulants (DOACs). We acknowledge the importance of this information for clinical practice, particularly about thrombosis recurrence and bleeding complications; however, most studies and RCT had a relatively short median follow-up period, about 6 months on average. We have acknowledged this limitation in our conclusions and suggest that it is a crucial area for future research. “Our review highlights critical gaps in the use of DOACs in USVT treatment that need addressing through further research. While DOACs have shown promising results in USVT patients, it is essential to emphasize the need for long-term studies to provide more robust data on efficacy and safety over time. Not only extend the duration but also delve into specific scenarios such as PVT related to hepatocellular carcinoma or catheter-related upper extremity thrombosis in cancer patients. Ongoing RCTs data will be crucial in developing tailored therapeutic strategies and guidelines to enhance patient outcomes in these complex conditions”.
The conclusion should highlight areas where further research is needed. For instance, the review could emphasize gaps in the literature, such as the scarcity of RCTs specifically focused on USVT or the need for more data on the long-term safety of DOACs in these patient populations. Future research directions could explore alternative treatment options, personalized therapy strategies, or the development of specific guidelines for DOAC use in uncommon venous thrombotic conditions. As you pointed out, there is indeed a need for more comprehensive studies on the long-term safety and efficacy of DOACs, as well as on specific venous thrombotic conditions.
We have addressed these gaps in the revised conclusion section of our manuscript, emphasizing the scarcity of randomized controlled trials and the urgent need for targeted research. “Our review highlights critical gaps in the use of DOACs in USVT treatment that need addressing through further research. While DOACs have shown promising results in USVT patients, it is essential to emphasize the need for long-term studies to provide more robust data on efficacy and safety over time. Not only extend the duration but also delve into specific scenarios such as PVT related to hepatocellular carcinoma or catheter-related upper extremity thrombosis in cancer patients. Ongoing RCTs data will be crucial in developing tailored therapeutic strategies and guidelines to enhance patient outcomes in these complex conditions”.

Reviewer 2 Report
Comments and Suggestions for Authors
Franco-Moreno et al. present a review of the safety and efficacy of the administration of DOAC for the treatment of unusual-site venous thrombosis (USVT). The rationale for the review was that while large investigations for common thrombotic and thromboembolic disorders have been performed assessing DOAC, there are investigations of relatively neglected clinical scenarios of thrombotic disorder that would benefit the readership.
USVT are defined by the authors as those disorders that involve thrombosis in atypical locations such as abdominal veins (splanchnic, renal, and ovarian), cerebral veins and venous dural sinuses, upper extremity venous system and retinal vein thrombosis. These target areas are depicted in figure 1.
The organizational approach to this narrative review is standard, with an informative description of the incidence and natural course of each thrombotic condition. Thereafter, accepted initial treatment of each condition is provided, and then the present state of affairs concerning DOAC administration for either short- or long-term treatment.
Depending on the condition, some RCT do exist with DOAC administration utilization. However, the deficits in investigation to date and lack of evidence is also noted. This is useful to the readership in identifying critical areas of needed clinical investigation and as a warning to not just simply administering DOAC.
Lastly, after a reasonable Conclusions section is presented, the authors present in table 2 the recommendations for anticoagulation for patients with rare thrombotic conditions as suggested by two societies and one large clinical trial.
In summary, this brief narrative review provides the readership with usable information to temper DOAC administration and identify weak and strong areas of evidence-based treatment in these rare settings. I have only one minor suggestion.
Comments
- Figure 1. Rather than have a picture of an intact body in the middle of the figure, the authors should place the incidence of thrombosis (%) next to each affected body part. Then the readership would be able to at one consideration put into perspective the rareness of each condition.
Author Response
Reviewer 2
Franco-Moreno et al. present a review of the safety and efficacy of the administration of DOAC for the treatment of unusual-site venous thrombosis (USVT). The rationale for the review was that while large investigations for common thrombotic and thromboembolic disorders have been performed assessing DOAC, there are investigations of relatively neglected clinical scenarios of thrombotic disorder that would benefit the readership.
USVT are defined by the authors as those disorders that involve thrombosis in atypical locations such as abdominal veins (splanchnic, renal, and ovarian), cerebral veins and venous dural sinuses, upper extremity venous system and retinal vein thrombosis. These target areas are depicted in figure 1.
The organizational approach to this narrative review is standard, with an informative description of the incidence and natural course of each thrombotic condition. Thereafter, accepted initial treatment of each condition is provided, and then the present state of affairs concerning DOAC administration for either short- or long-term treatment.
Depending on the condition, some RCT do exist with DOAC administration utilization. However, the deficits in investigation to date and lack of evidence is also noted. This is useful to the readership in identifying critical areas of needed clinical investigation and as a warning to not just simply administering DOAC.
Lastly, after a reasonable Conclusions section is presented, the authors present in table 2 the recommendations for anticoagulation for patients with rare thrombotic conditions as suggested by two societies and one large clinical trial.
In summary, this brief narrative review provides the readership with usable information to temper DOAC administration and identify weak and strong areas of evidence-based treatment in these rare settings. I have only one minor suggestion.
Comments
Figure 1. Rather than have a picture of an intact body in the middle of the figure, the authors should place the incidence of thrombosis (%) next to each affected body part. Then the readership would be able to at one consideration put into perspective the rareness of each condition.
We have added the incidence of thrombosis in Figure 1.

Round 2
Reviewer 1 Report
Comments and Suggestions for Authors
Authors addressed my comments well.